# Cultural and contextual adaptation of mental health measures in Kenya: An adolescent-centered transcultural adaptation of measures study

Vincent Nyongesa[1]*, Joseph Kathono[1,2], Shillah Mwaniga[2,3], Obadia Yator[1], Beatrice Madeghe[4], Sarah Kanana[2], Beatrice Amugune[5], Naomi Anyango[6], Darius Nyamai[2], Grace Nduku Wambua[7], Bruce Chorpita[8], Brandon A. Kohrt[9], Jill W. Ahs[10,11], Priscilla Idele[12], Liliana Carvajal[13,14], Manasi Kumar[1,15]

1 Department of Psychiatry, University of Nairobi, Nairobi, Kenya, 2 Nairobi Metropolitan Services, Nairobi, Kenya, 3 Vrije University, Amsterdam, Netherlands, 4 Department of Food and Nutrition Sciences, University of Nairobi, Nairobi, Kenya, 5 School of Pharmacy, University of Nairobi, Nairobi, Kenya, 6 Department of Mental Health, Ministry of Health, Kenya, 7 Department of Clinical, Neuro and Developmental Psychology, Amsterdam Public Health Research Institute, Vrije Universiteit Amsterdam, Amsterdam, The Netherlands, 8 University of California, Los Angeles, United States of America, 9 Division of Global Mental Health, Department of Psychiatry and Behavioral Science, The George Washington University, Washington, District of Columbia, United States of America, 10 Department of Neurobiology, Care Sciences and Society, Karolinska Institutet, Solna, Sweden, 11 Department of Health Care Sciences, Swedish Red Cross University College, Huddinge, Sweden, 12 UN Secretariat, New York, New York, United States of America, 13 Division of Data, Analytics, Planning and Monitoring, Data and Analytics Section, UNICEF, New York, New York, United States of America, 14 Department of Global Public Health, Karolinska Institutet, Stockholm, Sweden, 15 Brain and Mind Institute, Aga Khan University, Nairobi, Kenyau

* nyongesavincent@gmail.com

**Data Availability Statement:** Relevant focus group discussions, cognitive interview transcripts, and a

## Abstract

### Introduction

There is paucity of culturally adapted tools for assessing depression and anxiety in children and adolescents in low-and middle-income countries. This hinders early detection, provision of appropriate and culturally acceptable interventions. In a partnership with the University of Nairobi, Nairobi County, Kenyatta National Hospital, and UNICEF, a rapid cultural adaptation of three adolescent mental health scales was done, i.e., Revised Children's Anxiety and Depression Scale, Patient Health Questionnaire-9 and additional scales in the UNICEF mental health module for adolescents.

### Materials and methods

Using a qualitative approach, we explored adolescent participants' views on cultural acceptability, comprehensibility, relevance, and completeness of specific items in these tools through an adolescent-centered approach to understand their psychosocial needs, focusing on gender and age-differentiated nuances around expression of distress. Forty-two adolescents and 20 caregivers participated in the study carried out in two primary care centers where we conducted cognitive interviews and focused group discussions assessing mental health knowledge, literacy, access to services, community, and family-level stigma.

summary table are within the Supporting information files.

**Funding:** MK received Fogarty International Centre's Emerging global leadership award (grant no K43 TW010716-04). The funders had no role in study design, data collection and analysis, decision to publish, or preparation of the manuscript.

**Competing interests:** he authors have declared that no competing interests exist.

## Results

We reflect on process and findings of adaptations of the tools, including systematic identification of words adolescents did not understand in English and Kiswahili translations of these scales. Some translated words could not be understood and were not used in routine conversations. Response options were changed to increase comprehensibility; some statements were qualified by adding extra words to avoid ambiguity. Participants suggested alternative words that replaced difficult ones and arrived at culturally adapted tools.

## Discussion

Study noted difficult words, phrases, dynamics in understanding words translated from one language to another, and differences in comprehension in adolescents ages 10–19 years. There is a critical need to consider cultural adaptation of depression and anxiety tools for adolescents.

## Conclusion

Results informed a set of culturally adapted scales. The process was community-driven and adhered to the principles of cultural adaptation for assessment tools.

## Introduction

### Why is it important to consider cultural adaptation of mental health tools?

Assessment of prevalence of mental health issues among adolescents, evaluation of interventions, and determination of cost-effectiveness of programs in low and middle-income countries (LMICs) proves difficult due to the lack of culturally adapted and validated tools for child and adolescent mental health (CAMH) [1]. Several mental health aspects, such as perception of health and illness, help-seeking behavior, practitioner and patient attitudes, are impacted by cultural diversity [2]. Culture influences what is considered a problem, how it is understood, and the kind of practical solutions considered [3]. To generate more data on adolescent mental health in LMICs, there is a need to consider cross-cultural issues also due to the high level of ethno-diversity in these contexts [4]. Of great concern is that mental health tools developed for populations in high-income countries may fail to accurately assess and identify the mental health issues in LMICs [5], which calls for considering an adaptation of tools to fit the specificity of diverse contexts. Additionally, assessing a tool against a gold standard diagnostic interview is crucial for the cross-cultural application of screening the tool [5]. A single mental health term can be understood differently in different contexts; for example, literature has highlighted the variety of local idioms for describing distress [6]. For instance, in Nepal, it is described as "heart-mind" problems [7], "thinking too much" in Haiti [8], "kufungisisa" in Shona of Zimbabwe [9], "Kufikiria sana" in Kenya [10] and these show the necessity for cultural adaptations.

### Transcultural translation and adaptation, and multi-stakeholder involvement

Transcultural translation and adaptation (TTA) is recommended process of using existing tools in other cultures, languages, or geographical areas. Adapting an existing tool saves costs and time than developing a new tool [11,12]; it also reduces the complexity of creating a new

tool [13]. Culturally adapted tools should produce a reliable and valid tool that does not deviate from the original tool and should enable comparison of results found in other regions [13,14]. In addition, multi-stakeholder inquiries involving representatives from different groups in the society are crucial as they bring different views that help tailor the tools to meet specific needs of the target group [15] and ensure that the tools are locally suitable and applicable [1,16]. The public and global mental health fields emphasize this and recognize the need for linguistic and cultural adaptation of mental health tools [17,18]. Involvement of end users, community members, and lived experience representatives in making decisions during adaptations and letting experts make sure final changes and decisions are inclusive, helps maintain touch with the local realities [19].

Issues of stigma, discrimination, socioeconomic inequalities, coping, and resilience are critical to this process of cultural contextualization of commonly used mental health tools for adolescent health. In addition, culture is impacted by the broader context of social norms and social issues [2]. Within the UNICEF-led initiative of Measurement of Mental Health among Adolescents at the Population level (MMAP) [18], aside from moving the needle on measurement of mental health outcomes and related indicators, there is an impetus towards capacity building of local academic, health, and advocacy structures to develop training-of-trainer models to strengthen the capacity to support adolescents' mental health [18,20]. Adoption of this approach, connecting programmatic work to systematic evidence gathering is aligned with global recommendations to build capacity of mental health systems in resource constraint contexts [21].

Our objectives were to conduct focus group discussions (FGDs), and cognitive interviews (CIs) to develop an improved understanding of the culturally adapted items and their meaning for different age groups of boys and girls, pregnant adolescent girls, and caregivers of younger adolescents. We also explored age and gender differences around idioms used to express mental health difficulties and distress.

This paper describes the application of the transcultural translation and adaption [7] approaches to a selected set of items from the Revised Children's Anxiety and Depression Scale (RCADS) [22], UNICEF mental health module for adolescents, and the Patient Health Questionnaire (PHQ-9) set [23] tool in Nairobi, Kenya.

## Methods

### Settings and study sites

The study was conducted in two government-owned urban-based health care facility sites (Kariobangi and Kangemi) [24]. These centers provide non-specialized primary health care services, including Maternal Child Health Care, and are operated by a limited number of Nurses and clinicians. Both health care centers are level three facilities under the Nairobi Metropolitan Services (formerly known as Nairobi County Health Services). Level three facilities include health centers, maternity homes, and sub-district hospitals. Kariobangi health center is in a low-income residential area in the northeastern part of Nairobi, Kenya. It consists of the lower middle class and informal settlements with approximately 18,903 residents [25]. On the other hand, Kangemi Health Center is located in an informal settlement in Nairobi City within a small valley on the city's outskirts with approximately 116,710 residents [25]. The two study locations have similar characteristics: cosmopolitan, densely populated urban informal settlements. These areas have high drug abuse and crime levels coupled with youth unemployment and idleness. Other studies have demonstrated high prevalence of mental disorders in school-going children in Kenya [26], with substance abuse and depressive disorders being common [27]. These adolescent difficulties have been made worse by the COVID-19 pandemic [28,29].

## Participants

In identifying the study participants, non-probability purposive sampling targeted adolescent boys and girls living in low-resource settings. These participants were mobilized by trained community health volunteers (CHVs), who administered consent and assent a few days before the focus group discussions (FGDs) and cognitive interviews (CIs). Six FGDs were conducted–This was a moderator-guided discussion that involved participants with similar characteristics and experiences who responded to questions exploring specific topics of interest. Sixteen CIs were also conducted–Individual interviews whereby the participant responded to questions asked by the interviewer to describe an experience or viewpoint on a topic of interest. These FGDs and CIs were carried out in November and December 2020 among 62 participants.

## Study design

This qualitative study explored the cultural acceptability, comprehensibility, relevance, and completeness of items in three adolescent mental health tools- RCADS, PHQ-9, and UNICEF mental health module. The design also reflected the TTA approaches, with qualitative data reported according to the COREQ checklist [30] (S1 Checklist). The TTA process uses a series of systematic steps to assess an array of cultural equivalence domains [31]. In TTA, the tools were translated by bilingual experts, then reviewed by mental health experts. FGDs followed this, then CIs, while adopting any suggested changes in the wording of the tools. Finally, a back-translation was done to check whether the tools retained their initial meaning [32–34].

## Ethical clearance

The study was approved by the Kenyatta National Hospital/University of Nairobi ethical review committee (approval no. P694/09/2018). In addition, approval was received from Nairobi County Health no. CMO/NRB/OPR/VOL1/2019/04 and a permit from Kenyan National Commission for Science, Technology, and Innovation (NACOSTI/P/19/77705/28063) was obtained. We obtained assent from participants below 18 years old and consent from their parents or guardians.

## Focused group discussion and cognitive interviews

We conducted six focus group discussions (N = 46) among adolescents ages 10–19 years (n = 40) and caregivers to adolescents ages 10–14 years (n = 6) (See Table 1).

We also conducted cognitive interviews (n = 16) among twelve adolescents, including pregnant and parenting adolescents and four caregivers to adolescents ages 10–14 years.

**Table 1. Summary of FGD participants.**

| FGD set | Site | Cohort | N = 46 |
| --- | --- | --- | --- |
| First FGD | Kangemi health center | Girls 10–14 years | 8 |
| Second FGD | Kariobangi health center | Boys 10–14 years | 8 |
| Third FGD | Kariobangi health center | Girls 15–19 years | 8 |
| Fourth FGD | Kangemi health center | Boys 15–19 years | 6 |
| Fifth FGD | Kariobangi health center | Caregivers | 8 |
| Sixth FGD | Kangemi health center | Caregivers | 8 |

A table showing categories of participants and numbers for the different FGDs.

FGDs and Cis were the methods used to conduct transcultural translation and adaptation processes on an abbreviated version of the Revised Children's Anxiety and Depression Scale (RCADS) items covering the subscales of major depressive disorder, generalized anxiety disorder, separation anxiety disorders, social phobia, and panic disorder [22]. The RCADS is a widely used instrument for collecting information on depression and anxiety symptoms in children and adolescents. We also used items from the Patient Health Questionnaire (PHQ-9) set, a brief and widely used screening measure of depressive symptomology [23].

The FGDs included activities like body mapping to acclimatize and elicit some of the feelings in the different parts of the body under circumstances of sadness or happiness. In addition, understanding free-listed mental health terms and some of the idioms or colloquial words used were also explored. Subsequently, the participants were taken through English and Kiswahili versions of the tools to discuss various aspects of each element, following the TTA methods [31] established in the MMAP protocol. These domains are comprehensibility, acceptability, relevance, completeness, and relevance [18].

The cognitive interviews focused on participants' understanding of the specific wording of the tools. Each participant was either given an English tool or a Kiswahili tool and taken through each statement to gauge their comprehension and any problematic words identified and suggested alternative wordings provided by the participant.

See Box 1, which provides vital information on domains covered during the FGDs and Cis [1]. The FGDs also looked at cultural practices, understanding of mental health problems, associated service availability, and caregiver and adolescent recommendations on needed services.

## Box 1. Cross-cultural equivalence domains

| Domains covered | Focused group discussions (FGDs) | Cognitive interviews (Cis) |
|---|---|---|
| *Comprehension*–if the translation is understandable in a language known to the local population | Participants were asked to rephrase the statement or questions to evaluate their comprehension | Participants rephrased the statements in their own words to gauge their understanding |
| *Acceptability*–if other respondents would be uncomfortable responding honestly to the question or statement at hand | Participants' opinions were sought on whether peers would feel uncomfortable responding to any part of the statement and in case they wished for changes to accommodate | Participants were asked for their opinion if other adolescents of similar age as the one who was being interviewed would respond to the question without reservations |
| *Relevance*–if the question or statement is relevant to local culture | Participants were asked if the statements or questions represented daily issues within the society | Participants were asked the wordings were commonly used in their immediate surroundings by their peers |
| *Completeness*–if back-translation would relate to the same concepts and ideas as the original statement | Participants were given both English and Kiswahili versions of the tools and were able to check them and ensure the Kiswahili version would mean the same thing when translated back to English | Participants were subjected to one version of the tool; either English or Kiswahili, those who used Kiswahili version agreed that back-translation would make sense |

## Data collection and analysis

Sociodemographic data was collected on the day when focus group discussion and cognitive interview were conducted. Permission was sought from all participants to record the interviews, and each participant was identified by a number during the discussion for anonymity.

The study participants were taken through informed assent and consenting details to ensure they understood before signing the consent form. The consent highlighted the purpose of the study, benefits, risks, voluntary nature of participation, and withdrawal of consent at any stage of the study without being penalized. FGDs and Cis were facilitated by a team of 6 composed of female and male clinical psychologists and mental health researchers. All interviewers had prior training and field experience in conducting FGDs and Cis. Both the FGDs and Cis were carried out between November and December 2020 within the two health care facilities. Audio recordings were conducted following all protocols to ensure confidentiality and data protection.

The recordings were transcribed verbatim, and group members collated transcriptions during the process. Qualitative data from the sixteen cognitive interviews and the six FGDs were uploaded and analyzed in Nvivo version 10 Qualitative Data Analysis software [35]. Thorough reading through the content and identifying the texts and patterns linked to each theme were done. During this thematic content analysis, emerging themes were identified both deductively and inductively. Cross-tabulation and queries were used in analysis to compare the respondents' perspectives for each item in the PHQ-9 and RCAD tools. Participants transcribed responses to each statement, indicating if they understood or did not understand it. Therefore, this section was either coded 'participant understood or not understood,' which indicated comprehensibility during coding. Common patterns and discrepancies were identified during the process. In addition, adolescent experiences were also identified inductively and classified as independent themes.

## COVID-19-related adaptations for data collection and adolescent engagement

Working with adolescents and caregivers followed allCOVID-19 protocols set by the Kenyan Ministry of Health. A few facilitators were on the ground while others observed and participated via video conference using zoom or google meet set up for each FGD. In addition, we relied on our strong linkages with community health workers to make connections. During our data collection (November 2020- December 2020), Kenya experienced a strong first and second wave of COVID-19 infections surge. However, no participants or facilitators tested positive during this phase.

## Results

### Demographic characteristics

The average age of the adolescents who participated in the focus group discussions was 14 years (age range of 10–19 years). 46.7% were male adolescents, while 53.3% were female adolescents (See Table 2). The mean age for the adolescents who participated in the cognitive interviews was 14.9 years (age range of 10–18 years). 33.3% were male adolescents, while female adolescents were 66.7%. We also summarize characteristics of caregivers of adolescents aged 10–14 years who participated in the FGDs and Cis (See Table 2).

Our results are organized into segments:

a. we carried out FGDs to better understand the items and their meaning for different age groups, genders, and caregivers. In our sample, we also included pregnant and parenting adolescents to resonate with their experiences too,

b. we reflect on age and gender differences around idioms used to express mental health difficulties and distress.

**Table 2. Adolescent and caregiver demographic information.**

| | | Number of participants | | Percentage (%) | |
|---|---|---|---|---|---|
| | | Focus group discussions n = 30 | Cognitive interviews n = 16 | Focus group discussions | Cognitive interviews |
| **Adolescent participants** | | | | | |
| Age (Years) | 10–14 | 16 | 4 | 53.3 | 33.3 |
| | 15–19 | 14 | 8 | 46.7 | 66.7 |
| Gender | Male | 14 | 4 | 46.7 | 33.3 |
| | Female | 16 | 8 | 53.3 | 66.7 |
| Education level | Lower primary (class 3 and below) | 2 | 0 | 6.7 | 0 |
| | Upper primary (class 4–8) | 16 | 6 | 53.3 | 50 |
| | Secondary (form 1–4) | 10 | 5 | 33.3 | 41.7 |
| | Post-secondary | 2 | 1 | 6.7 | 8.3 |
| **Caregivers participants** | | n = 16 | n = 4 | % | % |
| Age (Years) | 30–34 | 2 | 1 | 12.5 | 25 |
| | 35–39 | 4 | 2 | 25 | 50 |
| | 40–44 | 6 | 1 | 37.5 | 25 |
| | 45–49 | 3 | 0 | 18.8 | 0 |
| | 50–55 | 1 | 0 | 6.2 | 0 |
| Gender | Male | 0 | 0 | 0 | 0 |
| | Female | 16 | 4 | 100 | 100 |
| Education level | Upper primary (class 4–8) | 3 | 1 | 18.7 | 25 |
| | Secondary (form 1–4) | 13 | 3 | 81.3 | 75 |

A table showing demographic information of adolescents and caregivers who participated.

Majority of the statements of our newly translated screening tools were comprehensible and contextually appropriate, a consensus that was arrived at after most participants rephrased the meanings of the statements well, especially during CIs. However, a few discrepancies were highlighted in the FGDs; we conducted a few cognitive interviews to check the changes suggested in FGDs and gather individual opinions. We tabulated a few problematic words during FGDs and CIs (See Table 3).

In the statement, 'I feel like I do not want to move,'–the word 'move' (*Kusonga/kusogea*) was not well-understood by some adolescent participants and a caregiver contextually. Its Swahili translation "*Kuendelea*" (Continue) was suggested in place of "*kusonga/kusogea*" (Move). The Swahili translation for "feeling" which was rendered as '*Kuhisi*' was not well understood, and instead, the word '*Kusikia*' (Kenyan direct English to Swahili translation) was suggested by both caregivers and the adolescent participants. '*Kusikia*' (hearing) is a word used mainly in Kenya to refer to 'feeling' and was easy for participants to understand.

Discussions about translated terminology in Kiswahili helped participants find more precise terminology for specific items. However, certain suggestions made by adolescents could not make sense in the sentences, which led to omitting them (also see S1 Table). Others made suggestions based on personal opinions and assumptions, which posed a challenge. For instance, "*Sina ladha ya kula chakula (I do not feel the taste of eating food)*" as an alternative for '*nina shida ya hamu ya chakula*' (I have problems with my appetite).

## Feedback from caregivers

Caregivers gave their feedback on wording based on their understanding and their child's understanding. However, their responses could be biased; the level of understanding could

**Table 3. Key findings from the cognitive interviews and focus group discussions about RCADS and PHQ9 items in English and Kiswahili translation.**

| Tools | Findings from FGDs | Findings from CI | Comprehension | Acceptability | Relevance | Completeness |
|---|---|---|---|---|---|---|
| **RCADS** | | | | | | |
| **I feel worried when I think someone is angry with me** *(nahisi/nasikia wasiwasi wakati ninapodhania mtu amenikasirikia)* | Was understood by all groups. | Caregiver thought "worried" would be difficult for younger adolescents and suggested replacing it with "afraid," or "scared," or "sad." | Comprehensible to FGD and CI participants | Acceptable by all participants | Relevant to local context | Complete |
| **Items including the words 'Suddenly" (***Ghafla***) and "for no reason" (***Bila sababu***) posed challenges, e.g., items AD13, AD21, AD28, AD30** | Younger boys ages 10–14 years had difficulty with the word "suddenly" but never gave an alternative word. The group that had adolescent boys aged 15–19 years suggested changing "when there is no reason for this" to "without a reason." | Participants suggested omitting the phrases "suddenly" *(Ghafla)*, "there is no reason" *(Bila sababu)* | Not comprehensible to some extent due to the phrase "for no reason" or "without a reason" | Not acceptable, since participants thought something happens for a reason | Not relevant | Incomplete |
| **I worry I might look foolish** *(nina wasiwasi ninaweza onekana mjinga)* | Was understood by all groups | An adolescent participant had difficulties understanding the word "foolish" | Comprehensible to all FGD participants and most CI participants, except one who could not comprehend the word "foolish" | Acceptable | Relevant | Complete |
| **I cannot think clearly** *(siwezi fikiria vizuri/ waziwazi)* | Was understood by all groups | A caregiver thought the word "clearly" would be difficult for young adolescents and suggested replacing it with the word "very well" | Comprehensible to all participants, except for a suggestion by a caregiver to simplify the phrase "very well" | Acceptable | Relevant in local context | Complete |
| **when I have a problem, I feel shaky** *(ninapokuwa na shida, /tatizo nahisi kutetemeka)* | The group with adolescent boys aged 15–19 years suggested replacing the word "shaky" with "tremble." | One adolescent had difficulties understanding the word "shaky" but never gave an alternative word | Comprehensible to all groups except for difficulty understanding the word "shaky" by one CI participant | Acceptable | Relevant | Complete |
| **I feel worthless** *(najihisi sina maana/thamani)* | Was understood by all groups | Word "worthless" was not understood by adolescents, and they suggested replacing it with "I am nothing." | Not comprehensible by most participants | Acceptable | Relevant | Not complete |
| **I am afraid of being in crowded places (like shopping centers, busy playgrounds, bus stations, busy streets, market places)** *(Naogopa kuwa mahali penye watu wengi (kwenye maduka makuu, sinema, kituo cha basi, uwanja wa michezo wenye shughuli nyingi))* | The group that had adolescent boys aged 15–19 years suggested the use of examples that can be understood by those who are in the villages | Was understood by all participants | Comprehensible | Acceptable | Could not be relevant, for example, to an adolescent in rural areas who does not know cinemas, supermarkets | Complete |

*(Continued)*

**Table 3.** (*Continued*)

| Tools | Findings from FGDs | Findings from CI | Comprehension | Acceptability | Relevance | Completeness |
|---|---|---|---|---|---|---|
| **I feel like I don't want to move** (*nahisi kama sitaki kusonga/kusogea au kutingishika*) | The group that had adolescent boys aged 15–19 years suggested removing the "*kutingishika*," the alternative Swahili word for "move." | Many participants found the Swahili word "*kusonga/kusogea*" (move) confusing. Suggested replacing it with "*kuendelea*" (continue). Another one interpreted the statement as "moving on with life or education." One of the caregivers also pointed out that the word "move" would be difficult for younger adolescents in this context | A bit difficult to comprehend | Acceptable | Not relevant in the Kiswahili version | Complete |
| **I feel afraid that I will embarrass myself in front of people** (*nahisi uoga kuwa nitajifanya nionekane mjinga mbele za watu*) | The group that had adolescent boys age 15–19 years suggested replacing Swahili words "*nitajifanya nionekane mjinga*" (I will embarrass myself) with "*nijajiaibisha*" (I will shame myself). | Participants understood it | Comprehensible | Acceptable | Relevant | Complete |
| **I would feel scared if I had to stay away from home overnight** (*ningehisi uwoga ikiwa itabidi nikae mbali na nyumbani usiku kucha*) | The group that had adolescent boys age 15–19 years suggested replacing Swahili words "*usiku kucha*" (overnight) with "*usiku wote*" (the whole night). | An adolescent suggested replacing "scared" with "sad." One of the caregivers felt that the Swahili word "*ningehisi*" (I would feel" was hard to understand, changing it to "*ningesikia*" made her understand, but that literally means "I would hear" | Comprehensible | Acceptable | Relevant | Complete |
| **I feel restless** (*nahisi sina utulivu*) | All groups understood | Some of the adolescents could not understand the word "restless." One of them suggested replacing it with the words "not comfortable" | Not comprehensible to some adolescents during CIs | Acceptable | Relevant | Complete |
| **PHQ9** | | | | | | |
| **Little interest or less happiness in daily activities** (*Kupoteza hamu au furaha katika shughuli za kila siku*) | All the groups understood the statement well | One adolescent could not understand the word "interest" but no suggestion of an alternative word | Not comprehensible to one adolescent during CI | Acceptable | Relevant | Complete |
| **Feeling bored, depressed, or hopeless** (*Kukosa furaha/kuboeka, mawazo mengi, au kukosa tumaini*) | All the groups understood the statement well | "Bored, depressed, hopeless" could not be well understood by our younger adolescents but never suggested alternative words | Not comprehensible during CIs to some adolescents | Acceptable | Relevant | Complete |

(*Continued*)

**Table 3.** (Continued)

| Tools | Findings from FGDs | Findings from CI | Comprehension | Acceptability | Relevance | Completeness |
|---|---|---|---|---|---|---|
| **Poor appetite or overeating** *(Kukosa hamu ya kula au kula sana)* | All the groups understood the statement well | One of our participants could not understand "poor appetite," while another one suggested replacing "Overeating" with "eating too much" | Comprehensible, except by one young adolescent | Acceptable | Relevant | Complete |
| **Feeling bad or as a failure about yourself or a disappointment to your family** *(Kuhisi vibaya au kwamba umeshindwa au umeaibisha familia yako)* | All the groups understood the statement well | One adolescent could not understand the word "Failure" but did not provide an alternative word | Comprehensible, except for one adolescent | Acceptable | Relevant | Complete |
| **Thoughts that you would prefer being dead, or of hurting yourself in some way** *(Mawazo kwamba ungependelea kufa, au kujiumiza)* | All the groups understood the statement well | One adolescent could not understand the word "Thoughts" but did not suggest an alternative word | Comprehensible, except for one adolescent | Not acceptable | Not relevant | Complete |

*Note*: *These are a few items that were hard for participants; most of the items were comprehensible. However, some discrepancies were identified in a few instances as indicated in the table. Some suggested wordings were also captured.*

vary from one child to another, despite the children being of the same age. For example, a caregiver of a child aged 14 years old could say that the child cannot understand a particular item, while a caregiver of a child aged 10 years thinks that her child understands the term; this depends on the level of exposure of the child to certain words, especially in English.

Response options were also discussed as part of FGDs and CIs; they included visual options illustrated by a glass of water diagrams for RCADs, and a "stone diagram" for PHQ9 was also used in the discussions. The RCADs terminologies: Never *(Sipati kabisa)* for an empty glass, sometimes *(Mara kwa mara)* ¼ glass, many times *(Mara nyingi)*/ ¾ glass, and all the time *(Kila wakati)* for a full glass. Adaptation of RCADs response options "always" to "all the time" and "often" to "many times" was informed by back-translation, which provided simplicity to the responses. The illustration of glass accompanied by words was preferred (see Fig 1 for this illustration) instead of using only words.

PHQ9 response categories reflect the number of days the participant feels bothered by a symptom over the previous two weeks. For instance, a visual aid represented the number of days in different stone sizes. For not at all *(hapana kabisa)*, the smallest "stone" was used, several days *(siku kadhaa)* bigger "stone," more than seven days *(Zaidi ya siku saba)* a much bigger "stone," and nearly every day *(Karibu kila siku)* the biggest "stone" was used (see Fig 2 for this illustration) [36]. Participants preferred response options with visuals and words. During tool testing, we added another illustration with calendar days (see Fig 3 for this illustration) and found [37] that participants liked it more than the "stone diagram."

## Age and gender differences around idioms used to express mental health difficulties and distress

There were age and gender differences in reported adolescent reactions to psychological disturbance. The older adolescents were more expressive, more objective with their sharing, and could give more detailed information about the lived experiences they shared as compared to

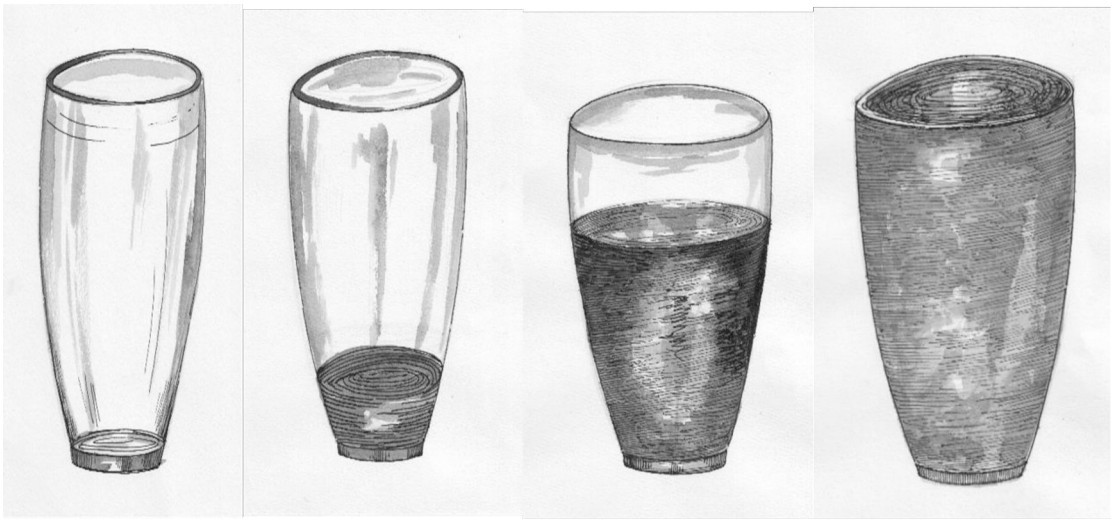

0 NEVER/SIPATI KABISA     1 SOMETIMES/MARA KWA MARA     2 MANY TIMES/MARA NYINGI     3 ALL THE TIME/KILA WAKATI

**Source:** *Glass diagram from* Kohrt et al., 2011 used for RCADS

**Fig 1. Illustration used to accompany word response options for RCADS.**

younger adolescents. They also understood more about the terminology used in mental health and showed more understanding of people undergoing mental health challenges as identified from free listed words checklist. On the other side, the younger adolescents took time to understand and respond to our questions. They also were less expressive and shared very few life experiences on mental health. There were gender differences in expressions too, as females

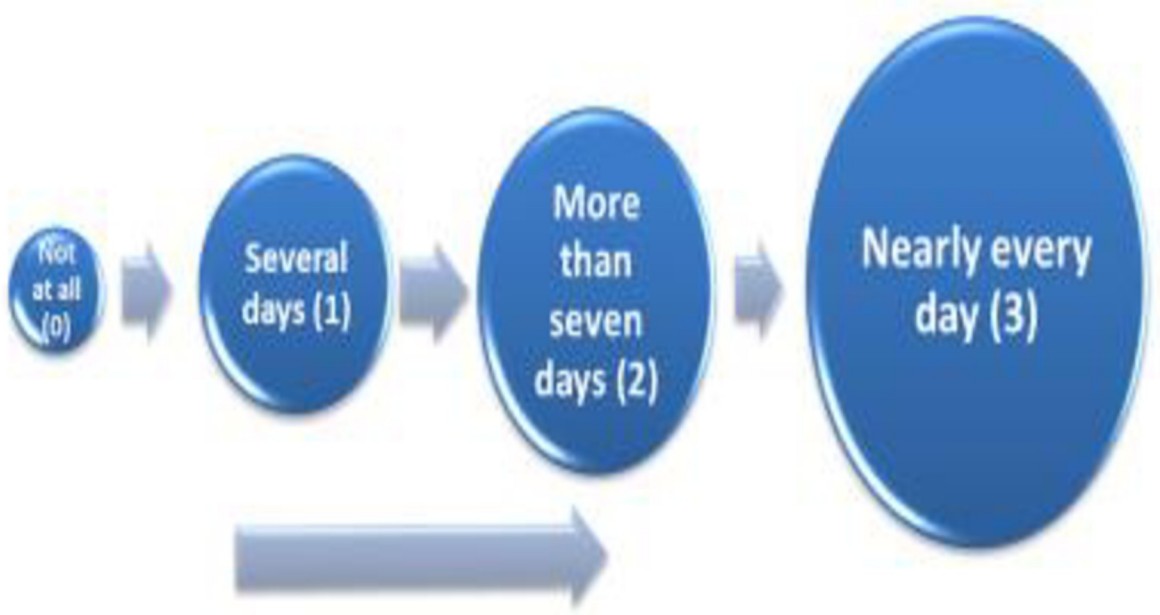

**Fig 2. Illustration used to accompany word response options PHQ9.**

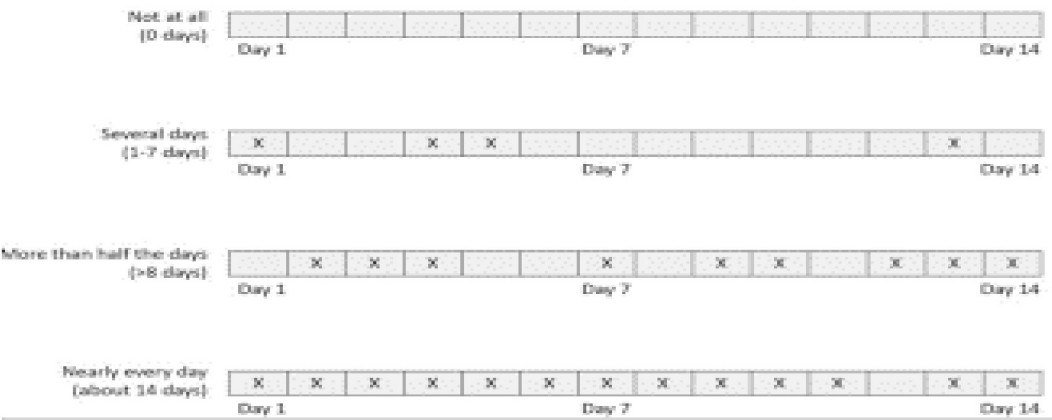

**Fig 3. Illustration used to accompany word response options PHQ9.**

expressed themselves more openly and took time to explain their views than male adolescents, who shared less and were briefer in content. The commonly mentioned reactions from body mapping exercise were anger, shaking, numbness, headache, urge to take alcohol, lack of sleep, appetite, socially withdrawn, inability to perform routine tasks, crying, and guilty feelings. In addition, more internalizing patterns were noticed in girls than boys. We felt that younger adolescents noticed more somatic and physical manifestations of distress, and their articulation got more expansive as we interviewed older adolescents. The impairment of functioning for adolescents with psychological disturbance was crosscutting in all ages and gender.

## Discussion

We conducted FGDs to develop an improved understanding of the RCADS, UNICEF mental health module for adolescents, and PHQ-9 items and their meaning for different age groups, both genders and including caregivers, pregnant and parenting adolescents. We identified words adolescents did not understand in these tools' English and Kiswahili translations. Some translated words could not be understood in their correct translation; instead, they were used in the routinely spoken format. Participants could not understand some of the response options; thus, they were changed to easily understood ones and embedded pictorials and worded response options. When translated to Kiswahili, some statements became ambiguous and thus were qualified by adding extra words to make more sense. We noticed that adolescents, in general, liked using the colloquial language, sheng, with older adolescents using sheng words that most younger adolescents could not understand. Sheng is a language spoken in the urban areas of Kenya, especially by adolescents, and it is a mixture of Swahili, English, and other Kenyan dialects from the diverse background of the urban inhabitants. Sheng terms used in one section of the city differ from what is found in another part of the city and are different from what is used in another town in Kenya. This made it challenging to adopt the colloquial sheng terminologies that were suggested.

## Implications for improved measurement and intervention development

Our findings reaffirm challenging experiences that young people go through that may contribute to the mental illness burden. Their lived experiences and those specific to mental distress need to be captured in a standardized manner which also includes appropriate contextualization so that whatever measurement these yield makes sense. We found that older adolescents better understood the tools in English, and we also learnt that our younger adolescents and caregiver participants preferred the use of Kiswahili for ease of understanding. Adolescents between the ages 15–19 years are mostly in secondary education, some completed, and others pursuing learning from vocational training institutions; thus, their understanding of English language is much better. We think this was the reason for older adolescents' understanding of English items to be better than younger adolescents. We benefitted from testing these tools in both languages since this exposed difficulty that crops up once a word has been translated from English to Kiswahili. This finding is similar to a cultural adaptation study carried out in Nigeria [38]. We found that the younger adolescents experienced greater challenges in grasping these constructs. However, our work also demonstrates the iterative nature of complex measurement tool adaptation and refinement of a cultural validation approach in piloting, revisiting, and refining the acceptability of various components of the screening tools during the piloting process, as highlighted in the MMAP protocol [18]. In many ways, our approach fostered a co-design model that may be suitable for further refining tools and interventions for young people [39]. Cultures vary with respect to the meanings they impart to illness and ways of making sense of the subjective experience of illness and distress [40]. The differences in cultures (including regional/sub-cultures, and cultures of mental health practitioners, the culture of youth) have a range of implications for words used in mental health practice. There are dual roles that participatory methods-driven cultural contextualization led by multiple stakeholders can play; such exercises improve the sensitivity and specificity of the tools. In addition, the culture of our youth, clinicians, and the service system also impacts outcomes and uptake of these measures [41]. In a recently published article presenting the impetus to the measurement of mental health among adolescents at the population level initiative, it was argued that *'you can't manage what you do not measure'* in the context of adolescent mental health [34]. How measurement is contextualized both developmentally and culturally is a critical consideration in understanding adolescent and youth mental health needs in diverse cultural contexts. Recently, the MMAP protocol covering four countries outlined the transcultural translation and adaptation process focusing on 9 step model that can be contextualized in suitable ways for country level adaptation [33]. Our work was embedded in this broad rubric.

During FGDs, participants reported understanding most of the words in English and Kiswahili; instead, during CIs, participants reported difficulty understanding more words from the same scales. The excellent understanding of wordings in FGDs could result from participants feeling pressure from peers to agree with the rest that they understand to avoid shame. CIs bring out the difficulties because it is a one-on-one interview that is considered more private; thus, there is no pressure from anyone, but just paraphrasing the words based on individual understanding. This highlights the need for conducting CIs after FGDs to develop more accurate wordings in the cultural translation process. We hope to build cultural competence through such exercises that underscore the recognition of adolescents' cultural understanding and then develop a set of skills, knowledge, and policies to deliver practical measurement tools and, ultimately, treatments [42].

We carried out a community-based process led by primary care facility-based health volunteers who helped identify participants. Youth responsive services are beginning to take a formalized shape in Kenyan primary care, but mental health within this range of services is still in

its nascent stage. Absence of appropriate tools, contextualized interventions, improved access, quality, and acceptability of these interventions matter. The range of difficulties that our inquiry exposed warrants in-depth service development and tangible referral pathways for a range of challenging life experiences. COVID-19 deepened insecurities and anxieties in the communities we worked with in this study. The adolescents and youth need more focused psychosocial, community, and educational support, which was deficient during the pandemic. Evidence also indicates that youth in informal settlements in Kenya are particularly impacted [43]. Peer support, improved service access, and effective self-management have been recommended [44], while caregiver mental health also needs to be on the table for improved outcomes for young people. In addition, psychological first aid and training of health care workers in understanding needs and offering simple self-management interventions have been recommended by the Kenyan Ministry of Health [45].

## Strengths and limitations

We sought views from young and older adolescents, and caregivers of younger adolescents to understand each group's perspectives. The adaptation process and wording of the culturally adapted tool was led by the users, which is a strength of the study. However, one clear limitation was that we conducted the study during the COVID-19 pandemic, which made it challenging to engage and interact with our participants closely and freely. We were also unable to get male caregivers since most of them were either at work or held up in other activities; thus, their views were not captured.

## Conclusion

The FGDs and CIs yielded meaningful information about RCADS, PHQ-9, and UNICEF mental health module (MMAP). We also gathered meaningful information around the cultural contextualization of these tools and a better understanding of mental health needs of adolescents and caregivers. The MMAP study protocol guided the cultural adaptation approaches for these tools for the adolescent population, and the participatory community-driven process was well-received. This process then led to an adaptation of the language and approach to assessment used for subsequent data collection and clinical validation.

## Supporting information

**S1 Checklist. COREQ checklist.**
(DOC)

**S1 Table. Item adjustment annex.**
(DOCX)

## Acknowledgments

Authors would like to thank other mentors in INSPIRE Kenya's work, adolescents and their caregivers who participated, and a fantastic team of community health workers and health facility workers of Kariobangi and Kangemi health centers for their support.

## Author Contributions

**Conceptualization:** Liliana Carvajal, Manasi Kumar.

**Data curation:** Vincent Nyongesa, Joseph Kathono, Shillah Mwaniga, Obadia Yator, Beatrice Madeghe.

**Formal analysis:** Beatrice Madeghe, Sarah Kanana, Grace Nduku Wambua, Manasi Kumar.

**Funding acquisition:** Liliana Carvajal.

**Investigation:** Joseph Kathono, Brandon A. Kohrt, Liliana Carvajal.

**Methodology:** Darius Nyamai, Grace Nduku Wambua, Brandon A. Kohrt, Liliana Carvajal, Manasi Kumar.

**Project administration:** Vincent Nyongesa, Joseph Kathono, Shillah Mwaniga, Beatrice Amugune, Naomi Anyango, Jill W. Ahs, Priscilla Idele.

**Resources:** Priscilla Idele.

**Supervision:** Obadia Yator, Sarah Kanana, Beatrice Amugune, Naomi Anyango, Jill W. Ahs, Priscilla Idele, Manasi Kumar.

**Validation:** Beatrice Madeghe, Beatrice Amugune, Darius Nyamai, Bruce Chorpita.

**Visualization:** Darius Nyamai.

**Writing – original draft:** Manasi Kumar.

**Writing – review & editing:** Vincent Nyongesa, Shillah Mwaniga, Obadia Yator, Beatrice Madeghe, Sarah Kanana, Beatrice Amugune, Naomi Anyango, Darius Nyamai, Bruce Chorpita, Brandon A. Kohrt, Jill W. Ahs, Priscilla Idele, Liliana Carvajal, Manasi Kumar.

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
