## [Decision Letter · Decision Letter 0]

20 Jul 2022

PONE-D-22-12888Cultural and contextual adaptation of mental health measures in Kenya: An adolescent-centered transcultural adaptation of measures studyPLOS ONE

Dear Dr. Kumar,

Thank you for submitting your manuscript to PLOS ONE. After careful consideration, we feel that it has merit but does not fully meet PLOS ONE’s publication criteria as it currently stands. Therefore, we invite you to submit a revised version of the manuscript that addresses the points raised during the review process.

We look forward to receiving your revised manuscript.

Kind regards,

Caroline Kingori

Academic Editor

PLOS ONE

Journal Requirements:

Reviewers' comments:

Reviewer's Responses to Questions

**Comments to the Author**

1. Is the manuscript technically sound, and do the data support the conclusions?

Reviewer #1: Partly

Reviewer #2: Partly

2. Has the statistical analysis been performed appropriately and rigorously? 

Reviewer #1: Yes

Reviewer #2: No

3. Have the authors made all data underlying the findings in their manuscript fully available?

Reviewer #1: Yes

Reviewer #2: Yes

4. Is the manuscript presented in an intelligible fashion and written in standard English?

Reviewer #1: Yes

Reviewer #2: Yes

5. Review Comments to the Author

Reviewer #1: • This was a very good and enjoyable manuscript. The need for culturally appropriate measurement scales is a logical point and well explained in the introduction section. My feedback is really on structure and writing to make sure it is readable and findings easy to review.

• Introduction section – provides a good overview of what the TAA process is and the importance of using a participatory method to improve measurement scales. In the objective, you mention only FGDs (no mention of C.Is).

• Box 1 – restructure/reformat table another way as the words seems to scatter off making it difficult to identify the 3 different columns (Domains, FGD & CD findings). Same applies for Table 3

• Page 31 – 3rd sentence “there are dual roles…” --- reconsider that sentence, reads a bit unclear

• Study implications are very well discussed though the latter discussion read a bit disconnected from the overall scope of the study. This last paragraph focuses, though accurate, the need for support services. The study findings tended to focus on the actual adaptation of the scales and not on the lived experience/challenges of the youth. Though important, in order for this to connect with this last paragraph, maybe providing some findings on that so that we can get context on what the service needs are.

• Writing – a few scattered grammatical errors noted (either missing a word or have an extra work somewhere. Be sure to review entire document to catch those hidden errors.

Reviewer #2: This study described the process of culturally adapting mental health assessment measures for adolescents in Kenya. This article is important because it highlights the lack of culturally tailored mental health tools for this population. The strength of this article is that authors used a thorough multi-stage adaptation process. To improve this article, I suggest that authors should focus on reducing some redundancy in the text and clarifying the methods that were used in the adaptation process.

Introduction

Page 6, Paragraph 1: This paragraph is quite redundant. The following four sentences say very similar things and can be combined.

• In addition, multi-stakeholder inquiries involving representatives from different groups in the society are crucial as they bring different views that help tailor the tools to meet specific needs of the target group (13).

• Global mental health emphasizes multistakeholder engagement and recognizes the need for linguistic and cultural adaptation of mental health tools (15,16).

• Therefore, involvement of service consumers or community members in making decisions during cultural adaptations and letting experts make sure final changes and decisions are inclusive helps maintain touch with the local realities (17).

• Checking with subject experts and end-users from time to time ensures relevance of the adapted tool.

Methods

• The paragraphs for study setting and study site should be combined

• It was hard to follow all of the data collection activities that were conducted and how they were ordered. To clarify this section, clearly define what the TTA approach is. Additionally, provide greater details about the protocols for each data collection activity .

• In analysis section, describe how data from body mapping and free listing activity were analyzed if differently from CI and FGD transcripts.

Results

• It is mentioned that pregnant and parenting adolescents were included as participants, however the extent to which their responses differed from non-parenting adolescents isn’t included in the results section.

• It was helpful to see the table with all of the adjustments that were made, but the table was hard to follow. This is partially due to formatting. It may be more useful to have that table as an appendix and include the final items in Kiswahili and English in the main text.

• Page 28-29: The section, “Age and gender differences around idioms used to express mental health difficulties and distress” is interesting, but it is unclear which parts of data collection the findings come from. It seems like a mixture of observations and the body mapping exercise. Additionally, parts of this section also are more appropriate for the discussion than the results section. For example, the sentences explaining what Sheng is should be in the discussion section.

Discussion

• The discussion notes that the study sought to gain an improved understanding of the “UNICEF mental health module for adolescents,” or “MMA protocol” yet that was not described in the methods and results. Include information in the methods describing what the module was, the methods for assessing the module, and participants assessment of the module.

• Include a paragraph where you describe any study limitations and strengths.

6. PLOS authors have the option to publish the peer review history of their article (what does this mean?). If published, this will include your full peer review and any attached files.

Reviewer #1: **Yes: **Elizabeth Wachira

Reviewer #2: No

---

## [Author Response · Author response to Decision Letter 0]

26 Sep 2022

30th August 2022

To

The Editor 

PLOS ONE

Re: Resubmission of paper ‘PONE-D-22-12888 Cultural and contextual adaptation of mental health measures in Kenya: an adolescent-centered transcultural adaptation of measures study’ 

Dear Dr. Caroline Kingori

We want to thank you for reviewing our manuscript and we have spent some time editing the paper and have provided point- by-point response given here below. We are grateful to the reviewers for their comments and feedback. 

We hope you the edited paper will meet your expectation. 

Regards 

Vincent Nyongesa 

Manasi Kumar 

Response to reviewers

Journal Requirements:

Response: we thank you and have now edited the manuscripts per guidance. 

Response: Thank you, we have added this statement (Please see page 9)

Response: we thank for noticing this inconsistency and we have now addressed this. The senior author was funded by Fogarty and the costs of activities were partially covered by UNICEF. 

Response: We have attached focus group discussion, and cognitive interview transcripts, and a table as supplementary information

Response: Thank you, we have corrected this now. 

5. Review Comments to the Author

Reviewer #1: • This was a very good and enjoyable manuscript. The need for culturally appropriate measurement scales is a logical point and well explained in the introduction section. My feedback is really on structure and writing to make sure it is readable and findings easy to review.

• Introduction section – provides a good overview of what the TAA process is and the importance of using a participatory method to improve measurement scales. In the objective, you mention only FGDs (no mention of C.Is).

Response: Thank you, we have now mentioned Cognitive interviews under this section

• Box 1 – restructure/reformat table another way as the words seems to scatter off making it difficult to identify the 3 different columns (Domains, FGD & CD findings). Same applies for Table 3

Response: Thank you, we have added table grids to make it readable

• Page 31 – 3rd sentence “there are dual roles…” --- reconsider that sentence, reads a bit unclear

Response: Thank you, we have cut down and restructured this sentence

• Study implications are very well discussed though the latter discussion read a bit disconnected from the overall scope of the study. This last paragraph focuses, though accurate, the need for support services. The study findings tended to focus on the actual adaptation of the scales and not on the lived experience/challenges of the youth. Though important, in order for this to connect with this last paragraph, maybe providing some findings on that so that we can get context on what the service needs are.

Response: Thank you for pointing at this, we have clarified this section (See page 32)

• Writing – a few scattered grammatical errors noted (either missing a word or have an extra work somewhere. Be sure to review entire document to catch those hidden errors.

Response: Thank you for pointing to this, we have checked this on Grammarly software and edited the work

Reviewer #2: This study described the process of culturally adapting mental health assessment measures for adolescents in Kenya. This article is important because it highlights the lack of culturally tailored mental health tools for this population. The strength of this article is that authors used a thorough multi-stage adaptation process. To improve this article, I suggest that authors should focus on reducing some redundancy in the text and clarifying the methods that were used in the adaptation process.

Introduction

Page 6, Paragraph 1: This paragraph is quite redundant. The following four sentences say very similar things and can be combined.

• In addition, multi-stakeholder inquiries involving representatives from different groups in the society are crucial as they bring different views that help tailor the tools to meet specific needs of the target group (13).

• Global mental health emphasizes multistakeholder engagement and recognizes the need for linguistic and cultural adaptation of mental health tools (15,16).

• Therefore, involvement of service consumers or community members in making decisions during cultural adaptations and letting experts make sure final changes and decisions are inclusive helps maintain touch with the local realities (17).

• Checking with subject experts and end-users from time to time ensures relevance of the adapted tool.

Methods

Response: Thank you, we have combined and edited the sentences per your suggestion (See page 6)

• The paragraphs for study setting and study site should be combined

Response: Thank you for pointing to this, we now have merged them (Please see page 7)

• It was hard to follow all of the data collection activities that were conducted and how they were ordered. To clarify this section, clearly define what the TTA approach is. Additionally, provide greater details about the protocols for each data collection activity.

Response: Thank you, we have elaborated the steps, and this work has been explained in the following publications we have now added more information here (Please see page 8). 

Joseph Hayes, Liliana Carvajal, Zeinab Hijazi, Jill Witney Ahs, P. Murali Doraiswamy, Fatima Azzahra El Azzouzi, Cameron Fox, Helen Herrman, Charlotte Petri Gornitzka, Brandon Staglin, Miranda Wolpert,You Can’t Manage What You Do Not Measure - Why Adolescent Mental Health Monitoring Matters, Journal of Adolescent Health, 2021, ISSN 1054-139X, https://doi.org/10.1016/j.jadohealth.2021.04.024.(https://www.sciencedirect.com/science/article/pii/S1054139X21002214)

Carvajal L, Ottman K, Ahs JW, Li GN, Simmons J, Chorpita B, Requejo JH, Kohrt BA. Translation and Adaptation of the Revised Children's Anxiety and Depression Scale: A Qualitative Study in Belize. J Adolesc Health. 2022 Aug 4:S1054-139X(22)00494-3. doi: 10.1016/j.jadohealth.2022.05.026. Epub ahead of print. PMID: 35934586.

Liliana Carvajal, Jill W. Ahs, Jennifer Harris Requejo, Christian Kieling, Andreas Lundin, Manasi Kumar, Nagendra P. Luitel, Marguerite Marlow, Sarah Skeen, Mark Tomlinson, Brandon A. Kohrt, Measurement of Mental Health Among Adolescents at the Population Level: A Multicountry Protocol for Adaptation and Validation of Mental Health Measures, Journal of Adolescent Health,2022,ISSN 1054-139X,https://doi.org/10.1016/j.jadohealth.2021.11.035. (https://www.sciencedirect.com/science/article/pii/S1054139X21006935)

• In analysis section, describe how data from body mapping and free listing activity were analyzed if differently from CI and FGD transcripts.

Response: Thank you, the data from body mapping and free listing were part of FGDs, thus analyzed together 

Results

• It is mentioned that pregnant and parenting adolescents were included as participants, however the extent to which their responses differed from non-parenting adolescents isn’t included in the results section.

Response: Thank you for pointing to this, however, their responses were not so different from the non-parenting ones, they expressed more problems that come with conceiving at a younger age and challenges associated with continuation of education. 

• It was helpful to see the table with all of the adjustments that were made, but the table was hard to follow. This is partially due to formatting. It may be more useful to have that table as an appendix and include the final items in Kiswahili and English in the main text.

Response: Thank you, we have added table grids for easy readability, we have also provided supplementary material with final wording. 

• Page 28-29: The section, “Age and gender differences around idioms used to express mental health difficulties and distress” is interesting, but it is unclear which parts of data collection the findings come from. It seems like a mixture of observations and the body mapping exercise. Additionally, parts of this section also are more appropriate for the discussion than the results section. For example, the sentences explaining what Sheng is should be in the discussion section.

Response: Thank you, we have now clarified this on page 29, we have also moved part of this paragraph to discussion per your suggestions

Discussion

• The discussion notes that the study sought to gain an improved understanding of the “UNICEF mental health module for adolescents,” or “MMA protocol” yet that was not described in the methods and results. Include information in the methods describing what the module was, the methods for assessing the module, and participants assessment of the module.

Response: the UNICEF publications came after the submission were made. We have now annotated those and reflected on some of these issues in the methods section. 

• Include a paragraph where you describe any study limitations and strengths.

Response: Thank you for this suggestion, we have included this section (See page 32)

---

## [Decision Letter · Decision Letter 1]

1 Nov 2022

Cultural and contextual adaptation of mental health measures in Kenya: An adolescent-centered transcultural adaptation of measures study

PONE-D-22-12888R1

Dear Dr. Kumar,

We’re pleased to inform you that your manuscript has been judged scientifically suitable for publication and will be formally accepted for publication once it meets all outstanding technical requirements.

Kind regards,

Caroline Kingori

Academic Editor

PLOS ONE

Additional Editor Comments (optional):

Authors address a topic of great concern in the public health arena. Mental health is still a challenge across the globe and does not receive adequate resources. I was glad to see authors discuss the importance of cross-cultural adaptation of mental health scales within a niche population of young people in Kenya. While many of the mental health scales readily used to measure burden and impact on the populace, cross-cultural adaptation of such scales within heterogeneous communities is not well studied. I commend the authors for taking on the task. I concur with the reviewers that the paper is ready for publication.

Reviewers' comments:

Reviewer's Responses to Questions

**Comments to the Author**

1. If the authors have adequately addressed your comments raised in a previous round of review and you feel that this manuscript is now acceptable for publication, you may indicate that here to bypass the “Comments to the Author” section, enter your conflict of interest statement in the “Confidential to Editor” section, and submit your "Accept" recommendation.

Reviewer #1: (No Response)

Reviewer #2: All comments have been addressed

2. Is the manuscript technically sound, and do the data support the conclusions?

Reviewer #1: (No Response)

Reviewer #2: (No Response)

3. Has the statistical analysis been performed appropriately and rigorously? 

Reviewer #1: (No Response)

Reviewer #2: (No Response)

4. Have the authors made all data underlying the findings in their manuscript fully available?

Reviewer #1: (No Response)

Reviewer #2: (No Response)

5. Is the manuscript presented in an intelligible fashion and written in standard English?

Reviewer #1: (No Response)

Reviewer #2: (No Response)

6. Review Comments to the Author

Reviewer #1: (No Response)

Reviewer #2: (No Response)

7. PLOS authors have the option to publish the peer review history of their article (what does this mean?). If published, this will include your full peer review and any attached files.

Reviewer #1: No

Reviewer #2: No

---

## [Editor Report · Acceptance letter]

1 Dec 2022

PONE-D-22-12888R1 

Cultural and contextual adaptation of mental health measures in Kenya: An adolescent-centered transcultural adaptation of measures study 

Dear Dr. Kumar:

I'm pleased to inform you that your manuscript has been deemed suitable for publication in PLOS ONE. Congratulations! Your manuscript is now with our production department. 

Kind regards, 

on behalf of

Dr. Caroline Kingori 

Academic Editor

PLOS ONE